# Tumor-Suppressor Role of the α1-Na/K-ATPase Signalosome in NASH Related Hepatocellular Carcinoma [note 1]

**DOI:** 10.3390/ijms23137359

**Published:** 2022-07-01

**Authors:** Utibe-Abasi S. Udoh, Moumita Banerjee, Pradeep K. Rajan, Juan D. Sanabria, Gary Smith, Mathew Schade, Jacqueline A. Sanabria, Yuto Nakafuku, Komal Sodhi, Sandrine V. Pierre, Joseph I. Shapiro, Juan R. Sanabria

**Affiliations:** 1Department of Surgery, Marshall University Joan C. Edwards School of Medicine, Huntington, WV 25701, USA; udohu@marshall.edu (U.-A.S.U.); banerjeem@marshall.edu (M.B.); rajan@marshall.edu (P.K.R.); sanabriaju@marshall.edu (J.D.S.); smith2152@marshall.edu (G.S.); schade4@marshall.edu (M.S.); sanabriaja@marshall.edu (J.A.S.); nakafuku@marshall.edu (Y.N.); sodhi@marshall.edu (K.S.); shapiroj@marshall.edu (J.I.S.); 2Marshall Institute for Interdisciplinary Research, Marshall University Joan C. Edwards School of Medicine, Huntington, WV 25703, USA; pierres@marshall.edu; 3Department of Nutrition and Metabolomic Core Facility, Case Western Reserve University School of Medicine, Cleveland, OH 44106, USA

**Keywords:** hepatocellular carcinoma, non-alcoholic steatohepatitis, α1-Na/K-ATPase, PI3K → Akt Pathway, FoxO3 signaling, pNaKtide

## Abstract

Hepatocellular carcinoma (HCC) is the second leading cause of cancer-related mortality worldwide, with an estimate of 0.84 million cases every year. In Western countries, because of the obesity epidemic, non-alcoholic steatohepatitis (NASH) has become the major cause of HCC. Intriguingly, the molecular mechanisms underlying tumorigenesis of HCC from NASH are largely unknown. We hypothesized that the growing uncoupled metabolism during NASH progression to HCC, manifested by lower cell redox status and an apoptotic ‘switch’ activity, follows a dysregulation of α1-Na/K-ATPase (NKA)/Src signalosome. Our results suggested that in NASH-related malignancy, α1-NKA signaling causes upregulation of the anti-apoptotic protein survivin and downregulation of the pro-apoptotic protein Smac/DIABLO via the activation of the PI3K → Akt pro-survival pathway with concomitant inhibition of the FoxO3 circuit, favoring cell division and primary liver carcinogenesis. Signalosome normalization using an inhibitory peptide resets apoptotic activity in malignant cells, with a significant decrease in tumor burden in vivo. Therefore, α1-NKA signalosome exercises in HCC the characteristic of a tumor suppressor, suggesting α1-NKA as a putative target for clinical therapy.

## 1. Introduction

Nonalcoholic fatty liver disease (NAFLD), including its inflammatory form, non-alcoholic steatohepatitis (NASH), is one of the manifestations of metabolic syndrome, which has become an overly prevalent condition in the western world, affecting up to 45% of the overweight population [1,2,3]. NASH can evolve to hepatocellular carcinoma (HCC), a highly lethal disease that constitutes 90–95% of primary hepatic cancers [1,4]. Nevertheless, the molecular mechanisms underlying this progression remain unclear [1,5,6,7,8]. Our group and others have previously reported the signaling function of the α1-subunit of Na/K-ATPase (NKA) [9,10,11,12,13,14,15], including recent reports on organogenesis during cell development [16]. Overactivation of the α1-NKA/Src signalosome by oxidative stress during pathological states results in the phosphorylation of Src kinase (pSrc), which activates downstream the PI3K → Akt → mTOR pathway [9,10,11,12,13,14,15,17,18,19,20,21]. Activation of the PI3K-Akt pathway promotes the expression of the anti-apoptotic protein survivin, driving abnormal cell growth, survival, proliferation, angiogenesis, and metabolism [20]. 

Survivin, an anti-apoptotic protein, regulates the cell cycle during the G2/M phase. It is essential during embryonic and fetal development, but absent in normal adult tissues [22,23]. In cancer, survivin is highly dysregulated and is present in all cell cycle phases, mainly in the cytoplasm, but also shuttles between the cytoplasm and nucleus via a CRM1/exportin-dependent pathway [24,25]. While its cytoplasmic pool inhibits apoptosis, its nuclear pool controls mitosis [25,26], and a third survivin cell pool (mitochondrial) is critical in cancer development and progression because of its higher anti-apoptotic effect when compared with the cytosolic pool [27,28,29]. Townley and Wheatley (2020) showed that survivin localizes in the mitochondria of malignant cells and its mitochondrial residence represents a gain of function over its physiological roles, driving cancer development and progression by reducing oxidative phosphorylation with greater dependency on glycolysis, changes that evoke the ‘Warburg effect’ [25]. Such a metabolic switch, critical for cancer cell development and survival [30], is also promoted by the activation of the proto-oncogene Src kinase [31,32]. Additionally, survivin has been shown to exert its anti-apoptotic function by delaying the release of the pro-apoptotic protein Smac/DIABLO (second mitochondria-derived activator of caspases/direct inhibitor of apoptosis-binding protein with low pI, or SMAC) from the mitochondria, as well as directly blocking SMAC cytosolic apoptotic activity [33,34,35]. The present study aimed to determine the molecular mechanisms that drive NASH-to-HCC and define the role of the α1-NKA signalosome in NASH progression. Our study revealed that during NASH-to-HCC progression, dysregulation of the α1-NKA signalosome upregulates survivin with concomitant downregulation of SMAC expression through the PI3K → Akt → S6K1 signaling pathway, favoring a change from programmed cell death to uncontrolled division. Normalization of the α1-NKA/Src signalosome by pNaKtide (developed from the α1-NKA subunit) results in an apoptotic “on switch” in malignant cells, leading to tumor regression.

## 2. Results 

### 2.1. Effect of α1-Na/K-ATPase Receptor Complex Normalization in Two Human HCC Cell Lines (Hep3B and SNU475)

α1-NKA signalosome normalization by pNaKtide induced a significant and dose-dependent increase in apoptotic activity in two human HCC cell lines when compared with untreated cells (Figure 1a and Appendix A), including cells that showed mitotic catastrophe (Figure 1b). Mitotic catastrophe is a type of cell death that occurs because of mitotic failure [36,37]. Morphologically (Figure 1b), this results in the formation of large cells with multiple micronuclei and decondensed chromatin [37]. Furthermore, cell viability analyses in the two cell lines (MTT assays) showed that the cytotoxic effect of pNaKtide at 50% inhibitory concentration (IC50 pNaKtide) was comparable to that of two known HCC chemo-active and FDA-approved drugs for clinical use (sorafenib and doxorubicin, Figure 1c). Interestingly, the IC50 of pNaKtide was 10-fold lower in SNU475 cells than in Hep3B cells (6 µM vs. 62.5 µM). SNU475 cells express significantly less α1-NKA, have lower pY260 α1-NKA (a marker for NKA signalosome) [32] and a markedly higher pY418 Src than Hep3B cells (Appendix A). These findings on apoptosis confirmed our previous report on malignant cell growth arrest by pNaKtide [17].

### 2.2. Effect of α1-Na/K-ATPase Receptor Complex Normalization on α1-Subunit, Src/ERK Kinases and Survivin/SMAC Protein Expressions in Two Human HCC Cell Lines

IC50-pNaKtide treatment of Hep3B and SNU475 cells significantly upregulated α1-subunit expression in comparison with the untreated cells. An increase in expression was observed as early as 2 h and up to 24 h (Figure 1d). In contrast, there was significant downregulation of pSrc in the Hep3B cell line, with only a trend in the SNU475 cell line (Figure 2a). Additionally, pERK was significantly downregulated in both cell lines, an effect that lasted up to 24 h (Figure 2a). Concomitantly, a significant upregulation of survivin and downregulation of SMAC protein were noted in the untreated human HCC cell lines (Figure 2b), and pNaKtide treatment influenced the expression of both proteins with attenuation of survivin and heightening of SMAC in a dose-dependent manner (Figure 2b and Appendix A). 

### 2.3. Effect of α1-Na/K-ATPase Receptor Complex Normalization on Apoptosis in Liver and Tumor Cells from Mouse Models of NASH & NASH-HCC

In the NASH model, we observed a paucity of apoptosis in the pNaKtide-treated animals when compared with untreated animals at 24 weeks using the TUNEL assay (on-to-off apoptotic switch, Figure 3a and Appendix A). Preclinical and clinical studies have positively correlated apoptotic activity with NASH progression [38], which confirmed the results of our earlier studies [39]. In contrast, NASH-HCC animals treated with pNaKtide showed a remarkable increase in tumor apoptotic activity compared with non-treated animals at both 24 and 28 weeks (off-to-on apoptotic switch, Figure 3a and Appendix A). Tumor burden analysis on H&E slides, where malignant vs. normal parenchymal cells were masked by color-pixel attributed channels from treated vs. untreated animals, showed a significantly lower tumor load in a dose-dependent manner at 24 weeks (early tumor) and 28 weeks (advanced tumor, Figure 3a and Appendix A). 

### 2.4. Effect of α1-Na/K-ATPase Receptor Complex Normalization in the SCID Mice Xenografts Tumors

Normalization of α1-Na/K-ATPase signalosome by pNaKtide in HCC xenograft tumor tissues of SCID mice revealed distinct histological differences (as shown by H&E staining, Figure 4a and Appendix A) between the treated and untreated mice. Xenograft HCC tumor size assessment via IVIS (video imaging) and electronic caliper measurements showed a significant decrease in tumor size between pNaKtide-treated and -untreated mice (Figure 4a and Appendix A), with concomitant heightened apoptosis in the pNaKtide-treated group compared with that in the untreated mice (*off-to-on* apoptotic switch*,*
Figure 4a and Appendix A). Additionally, normalization of the α1-Na/K-ATPase signalosome caused an upregulation of the α1-subunit expression in the xenograft tumors of the treated mice when compared with the untreated mice (Figure 4b). In contrast, we observed a significant increase in survivin expression in the untreated xenograft tumor cells and downregulation in tumor tissues of the pNaKtide-treated mice (Figure 4c and Appendix A). These findings reproduced our in-vitro data in the HCC cell lines (Figure 1a,d and Figure 2b). 

### 2.5. Effects of α1-Na/K-ATPase Receptor Complex Normalization in Mouse Models of NASH & NASH-HCC

Baseline upregulation of survivin expression was significantly attenuated in the livers of pNaKtide-treated animals in both animal models (NASH and NASH-HCC; Figure 5 and Appendix A). Concomitantly, the baseline low expression of SMAC significantly increased in treated vs. non-treated livers (Figure 5 and Appendix A). Survivin plays a critical role in the release of SMAC from the mitochondria and in the inhibition of downstream caspases following apoptotic stimuli [33,35]. Upregulation of survivin, as we observed in HCC tumor cells, causes a delay in the release of SMAC from the mitochondria mediated by a simple size-exclusion mechanism after its direct mitochondrial binding with SMAC [33,34]. Furthermore, survivin regulates the activity of cytosolic SMAC by neutralizing SMAC’s effect on the pro-apoptotic proteins cascade [33,35].

### 2.6. Survivin and SMAC Proteins Expression in Human Subjects

Findings in human livers reconciled the in-vitro and in-vivo studies, where survivin was highly expressed in the livers of patients with HCC as compared with normal subjects, patients with NASH, or patients with liver metastasis. In contrast, we observed a significant reduction in SMAC expression in the liver tissue of patients with NASH, HCC, and liver metastasis when compared with normal subjects (Figure 5 and Appendix A).

### 2.7. Signalosome of the α1-NKA

RNA sequencing of untreated human HCC cell lines (Hep3B and SNU475) revealed activation of the PI3K → Akt → mTOR → S6K1 pathway associated with inhibition of pro-apoptotic FoxO3 signaling (Figure 6a–f, Figure 7 and Appendix A). We observed an increased pAkt and pS6k1 state, confirming signaling activity through the PI3K → Akt pathway in human HCC cell lines (Figure 6c), and progressive reduction in kinases phosphorylation in cells exposed to pNaKtide, indicating an association of the pathway with α1-NKA (Figure 6c). Furthermore, the PI3K inhibitor wortmannin significantly decreased cell proliferation in Hep3B cell lines (Figure 6d) by downregulating survivin expression, the most potent anti-apoptotic protein that has been discovered to date [40]. To further determine whether the circuit that increased survivin expression observed in the HCC cell lines originated at the α1-NKA signalosome via PI3K → Akt axis activation/amplification, α1-NKA knockdown (KND) cells were generated from the two human HCC cell lines using small interfering RNA (siRNA). The α1-NKA-KND cell line significantly overexpressed survivin (Figure 6e). Additionally, digoxin, a specific ligand of NKA, was used to test the effect of α1-NKA signalosome activation on survivin expression. Digoxin significantly upregulated survivin expression in a dose-dependent manner, an effect that was rescinded in cells treated with pNaKtide (Figure 6f). Digoxin mediates its action by binding to the α-subunit of NKA and is the only FDA-approved cardiac glycoside indicated for the treatment of patients with mild or moderate heart failure and reduced ejection fraction [41,42]. Additionally, the administration of pNaKtide enhanced FoxO3 apoptotic signaling in the two pNaKtide-treated HCC cell lines by significantly increasing the nuclear expression of FoxO3 (Figure 7 and Appendix A). The forkhead box O (FoxO) family of transcription factors is recognized as a tumor suppressor that plays key roles in cell cycle arrest, senescence, apoptosis, differentiation, DNA damage repair, and scavenging of reactive oxygen intermediates (ROI) [43,44].

## 3. Discussion

Our findings showed that dysregulation of the α1-Na/K-ATPase signalosome in NASH-related HCC is associated with concomitant downregulation of the α1-NKA subunit, upregulation of the anti-apoptotic protein survivin, and downregulation of the pro-apoptotic SMAC protein expression, promoting a cell fate ‘switch’ from apoptosis to mitosis, which drives cell proliferation and tumorigenesis. Additionally, normalization of the α1-NKA signaling complex by pNaKtide reversed the expression of survivin and SMAC proteins to their physiological levels and enhanced apoptosis. Our data are in agreement with earlier reports of decreased α1-NKA expression, which has been implicated in other malignancies, such as prostate, pancreatic, and renal. [17,45,46]. Furthermore, our set of in-vivo studies showed that α1-Na/K-ATPase signalosome normalization reversed the “oncogenic apoptotic switch” that occurs during the progression of NASH-HCC, from *off to on*, inducing apoptosis in tumor cells of NASH-HCC murine models as well as in xenograft tumors of SCID mice with a corresponding decrease in tumor burden. Intriguingly, pNaKtide induced tumor cell apoptosis and tumor regression in the liver tissue and xenograft tumors of SCID mice, followed a by downregulation of survivin and concomitant upregulation of SMAC proteins in these mouse models. The interplay between survivin and SMAC proteins is critical for the regulation of cellular apoptosis. As an anti-apoptotic protein, survivin interacts with SMAC and neutralizes its inhibitory action on inhibitor of apoptosis proteins (IAPs), resulting in attenuation of caspase activity and abrogation of apoptosis [35,47,48]. Another important action of survivin, which may drive carcinogenesis, is the promotion of cell proliferation. During the G2/M phase of the cell cycle, survivin is highly expressed and binds to microtubules that make up the mitotic spindles. Such binding stabilizes the structure of the microtubules and prevents hydrolysis of the spindles, securing the integrity of mitotic organelles, evasion of growth arrest checkpoints, and assuring continuous cell division [48,49,50]. Additionally, survivin plays a key role in tumor angiogenesis, promoting the proliferation and migration of endothelial cells, and enhancing cancer cell survival [48,50,51]. Interestingly, the observed protein expression in both our in vitro and in vivo data, including our SCID mouse xenograft model, was recapitulated in the livers of patients with HCC compared with normal subjects, wherein we observed overexpression of survivin and low levels of SMAC protein in the livers of HCC patients compared with normal subjects (Figure 5). This finding suggests that survivin is a key protein that drives the progression of NASH to HCC. 

We further hypothesized that biological signals arising from the dysregulation of the α1-Na/K-ATPase signalosome following NASH-HCC progression are transmitted via the PI3K → Akt pro-survival pathway, directing an “oncogenic apoptotic” switch that favors cancer development and progression. The PI3K → Akt pathway has been reported to be constitutively activated in many types of malignancy. Factors that activate this pathway include loss of the tumor suppressor PTEN (phosphatase and tensin homolog deleted on chromosome 10), activation/amplification of PI3K → Akt, activation of growth factor receptors including epidermal growth factor receptor (EGFR), and exposure to carcinogenic agents [19]. PI3K → Akt signaling induces survivin transcription through phosphorylation of p70S6K1, driving cell proliferation, survival, and angiogenesis [19,20,40,52,53]. Our data suggest that normalizing the α1-NKA signaling complex with pNaKtide in cancer cells inhibits the PI3K → Akt pathway and downregulates survivin overexpression, resulting in malignant cell death and tumor regression. 

Furthermore, previous studies have shown that FoxO3 protein is phosphorylated during activation of the PI3K → Akt pathway prior to its extrusion from the nucleus to the cytoplasm, resulting in decreased expression of pro-apoptotic FoxO3 target genes, thereby favoring cancer initiation and progression [54]. Therefore, we assessed the nuclear-to-cytoplasmic export of FoxO3 transcription factor, following the dysregulation of the α1-Na/K-ATPase signalosome and its normalization by pNaKtide. Strikingly, our data revealed a significant increase in the cytoplasmic expression of FoxO3 (Figure 7) following the dysregulation of the α1-Na/K-ATPase signalosome and subsequent activation of the PI3K → Akt cascade in the untreated cancer cells, indicating an upregulation in the nuclear to cytoplasmic export of the FoxO3 protein. Interestingly, pNaKtide attenuated the nuclear-to-cytoplasmic export of FoxO3 protein and favored its nuclear localization (Figure 7). FoxO3 has been implicated as a tumor suppressor in HCC carcinogenesis [55] and *PTEN*-mediated tumor suppression is driven by the repression of survivin gene transcription via the direct binding of FOXO3 to the survivin promoter [56]. Additionally, previous studies have shown that FoxO3-induced apoptosis was enhanced upon survivin knockdown in cells [54,57] and the inhibition of the PI3K → Akt pathway resulted in the translocation of FoxO3 to the nucleus, leading to survivin repression. Furthermore, earlier studies have revealed that the activation of PI3K/Akt pathway phosphorylates FOXO3 and creates docking site for 14-3-3 proteins. The binding of 14-3-3 to FOXO3 results in nuclear export of FOXO3 [58]. Our findings suggest that the administration of pNaKtide reverses this phenomenon via the inhibition of the PI3K/AKT pathway, leading to the increase of nuclear localization of FOXO3.

Understanding the molecular mechanisms underlying the development and evolution of NASH-related HCC could shed light on checkpoint therapies for tumor regression. Our findings implicate dysregulation of the α1-NKA signalosome as the initiator of a signaling cascade, which through the PI3K → Akt pathway determines cell fate from programmed death to uncontrolled cell division, by a concomitant upregulation of survivin, downregulation of SMAC, and inhibition of the pro-apoptotic FoxO3 cascade. A progressive disarray of the α1-NKA signalosome that leads to NAFLD progression to NASH and concludes with the signaling of HCC genesis has been proposed (Figure 8). The signalosome is activated by a low cell redox status, widening a spiral cascade of metabolic disturbances that set cellular progression from wild-type metabolism to a stage of accelerated cell senescence to an on-apoptosis cell state (off-to-on switch). Cell progression in the continuous ROI feed-forward loop at α1-NKA promotes nuclear epigenetic changes and survivin overexpression, leading to uncontrolled cell division (on-to-off switch). In addition, α1-NKA signalosome normalization by pNaKtide not only prevents and reverses NASH-related metabolic changes, but its inhibitory effects on the PI3k → Akt pathway picture the signalosome as a tumor suppressor in NASH-related HCC. 

## 4. Materials and Methods

### 4.1. In-Vitro Studies

Human HCC cell lines (*Hep3B and SNU475* from ATCC, Cambridge, MA, USA) were tested for Mycoplasma (MycoAlert PLUS Mycoplasma Detection Kit, Lonza, Rockland, ME, USA) while growing in culture with high-glucose DMEM medium supplemented with 10% FBS and 1% penicillin/streptomycin (Hep3B), or with RPMI 1640 media supplemented with 10% heat-inactivated FBS and 1% penicillin/streptomycin (SNU475) in a 37 °C humidified incubator in the presence of CO_2_ at 5%. Cells were transiently transfected with an siRNA (small interfering RNA) α1-specific polypeptide (sense:5′-tcgagggtcgtctgatctttgatattcaagagatatcaaagatcagacgaccttttt-3′, and anti-sense: 5-ctagaaaaaggtcgtctgatctttgatatctcttgaatatcaaagatcagacgaccc-3) as previously described [11] to generate α1-NKA knockdown (KND) HCC cell lines and cultured as previously described. Cryopreserved primary human hepatocytes to serve as controls were obtained from Life Net Health (Virginia Beach, VA, USA) and cultured according to the supplier’s instructions using LifeNet Human Hepatocytes Culture and Supplement media (MED-HHCM-500ML and MED-HHPMS-15ML). The MTT cell proliferation assay was performed by plating 5000 cells/well in 6 wells (per condition) of 96-well plates and allowing them to adhere for 24 h, when media was replaced with media ± treatment agent for another 24 h to assess cell proliferation according to the manufacturer’s instructions (MTT assay, from ATCC^®^ #30-1010K, Cambridge, MA, USA). Digoxin (Sigma, St. Louis, MO, USA) was dissolved in serum-free media before addition to the cells at the indicated concentrations. 

### 4.2. NASH and NASH Related HCC Mouse Models

Seven-week-old female C57BL/6J mice (Jackson Laboratory, Farmington, CT, USA) were housed in a 12 h:12 h light-dark cycle in a temperature- and humidity-controlled environment. Following acclimation, mice were fed a standard mouse chow (NMC, Bio-Serv, NJ, USA) or a Western diet consisting of a high-fat diet (HFD, Bio-Serv, NJ, 60% of calories from fat) supplemented with 55% fructose-in-water ad libitum for 12 weeks. Mice typically develop NASH after 12 weeks with no visible tumors [59]. The NASH-HCC mouse model (STAM™ mice, Tokyo, Japan) was generated by injecting Streptozotocin (200µg STZ, Sigma, MO, USA) to neonatal male C57BL/6J mice 2 days after birth, and after 4 weeks of age, injected animals were exposed to a high-fat diet (HFD32, CLEA Japan, Inc., Tokyo, Japan) ad libitum. Mice typically develop NASH at 12 weeks, with the presence of HCC consistently by 16–20 weeks of age [60,61]. *Experimental Design.* After 12 weeks, mice were randomized into control and treatment groups (*n* = 7 per group), and the study continued for an additional 12 to 16 weeks as follows: (1) HFD with no treatment and (2) HFD treated with pNaKtide. While NASH female animals were treated with a fixed dose of pNaKtide (25 mg/kg BW dissolved in 100 µL 0.9% normal saline (NS), i.p. once a week)*,* the NASH-HCC male animals were treated with pNaKtide at a low (2 mg/kg BW X3 a week) or high dose (10 mg/kg BW X3 per week). All mice continued HFD ad libitum throughout the experimental period, which lasted 24 weeks for the NASH and early-stage HCC arm (12 weeks of treatment), or 28 weeks for the late-stage NASH-HCC arm (16 weeks of treatment; see animal flow chart, Appendix A). The animals were euthanized at the end of the study for liver and blood collection. The livers were washed with 0.9% normal saline at room temperature and divided into two before being snap-frozen in liquid nitrogen and stored at −80 °C or fixed at 4 °C (10% formaldehyde). The animals were subjected to all procedures according to the University IACUC-approved protocols.

### 4.3. Tumor Xenograft Model

To circumvent the lack of microenvironment in human HCC cell lines, we implanted a human HCC cell line (Hep3B) into seven-week-old female NOD SCID mice (NOD.CB17-Prkdcscid/J, JAX stock #001303, Jackson Laboratory, Farmington, CT, USA). Human HCC cells (Hep3B × 10^6^ cells) were implanted subcutaneously into each animal’s flank/back. NOD (CB17-Prkdcscid/J) is an immune-deficient mouse model lacking functional T and B cells, which makes this model suitable for xenogeneic tumor-graft experiments [62]. In addition, although some models may spontaneously develop partial immune reactivity (i.e., leakiness), this model is low on the leakiness scale, making it suitable for survival studies involving malignant xenograft cells [62]. The implanted cells were transfected with dual-luciferase (firefly luciferase) and fluorescent (mCherry) plasmid vectors using lentivirus to confer luminescence to the tumor for in vivo imaging using the IVIS system [63]. The tumor response to systemic 0.9% normal saline (NS) vs. pNaKtide in NS (30 mg/kg BW/week, in three doses) was video-monitored weekly. Animals were euthanized at eight weeks or when the tumor reached 1500 mm^3^ (to avoid animal suffering), and tumor/liver tissues and plasma were collected for assessment of tumor burden, apoptotic activity, and protein expression. 

### 4.4. Human Liver Tissues

Liver tissue samples were obtained from a tissue bank under IRB-approved protocol. Liver tissues were derived from surgical procedures at our institution from subjects with normal livers (*n* = 7), NASH (*n* = 17), HCC (*n* = 11), and liver metastases (*n* = 10) over a period of 3 years. Routine processing and evaluation of liver tissues were performed by experienced pathologists at our institution.

### 4.5. Treatment Agents

(i) *pNaKtide.* pNaKtide is a 33 amino acid peptide that prevents overactivation of α1 NKA/Src signalosome [64]. The sequence at the N domain of the α1-subunit of NKA that interfaces with the Src kinase domain was identified and subsequently synthetized (NaKtide = 20amino acids) and merged with a TAT leader sequence (13 amino acids) to establish cell permeability (pNaKtide, HD Bioscience Co Limited, Shanghai, China,. Consistently, pNaKtide prevents the formation of the NKA/Src receptor complex, interfering with the Src phosphorylation process, but does not affect Src activity regulated by IGF1. NKA interacts with Src through two binding motifs, namely the CD2 of the α1 subunit with the Src SH_2_ domain and the third cytosolic domain (CD3) of the α1 subunit with the Src kinase domain [17,64]. pNaKtide, which does not affect the ion-pumping function of NKA [64], appears to function as a Src regulator, mimicking the conformation-dependent scaffolding function of NKA [17]. It has been established that the CD3-Src kinase binding maintains Src in its inactive state, whereas in pathological states or when ouabain binds to NKA, Src kinase is released, activating different signaling pathways including ERK cascades, the PLC/PKC pathway, and mitochondrial production of reactive oxygen intermediates (ROI) [64,65]. pNaKtide was synthesized by mimicking the active sequence of the binding capacity of the α1-subunit of transmembrane NKA to Src (GRKKRRQRRRPPQSATWLALS RIAGLCNRAVFQTFA; MW 3877.64 Salt, HD Biosciences, San Diego, CA, USA). The compound is pure at >96% (LC-MS/MS) and was administered in a solution of normal saline (NS) (0.9%). A “scrambled” peptide (TAT-scrambled peptide: GRKKRRQRRRPPQACWIQNLSRSAGATVRLFLA) was used as the negative control for pNaKtide. (ii) *Sorafenib* (Bayer AG, Leverkusen, Germany) is an oral multikinase inhibitor approved by the FDA for clinical use in advanced-stage HCC. Since 2007, it has been recognized as a therapeutic option for patients with advanced unresectable HCC [66,67]. (iii) *Doxorubicin* (Pfizer, New York, NY, USA) is an anthracycline and one of the most commonly used agents in transarterial chemoembolization (TACE) for HCC patients at the intermediate stage [68]. (iv) *Digoxin* (Sigma-Aldrich, St. Louis, MO, USA) is a specific inhibitor of NKA [41,42] and is the only FDA-approved cardiac glycoside for the treatment of mild or moderate heart failure [41]. (v) *PP2* (Sigma-Aldrich, St. Louis, MO, USA) is a potent selective inhibitor of Src family kinases. [69] (vi) *AG490* (Tocris, Minneapolis, MN, USA) is a selective inhibitor of Janus kinase 2 (JAK2), which is a signal transducer and activator of the transcription 3 (STAT3) signaling pathway. [70] (vii) *Wortmannin* (Selleckchem, Houston, TX, USA) is a potent and selective inhibitor of PI3K protein [71].

### 4.6. Liver Apoptotic Activity Assessment

Apoptotic activity was assessed using the TUNEL (terminal deoxynucleotidyl transferase (TdT)-mediated dUTP nick end labeling) method for liver and tumor tissues and cell lines (Click-iT #C10617, Thermo Fisher Scientific, MA) following the manufacturer’s instructions. While cell line images were taken at 10× magnification, animal liver and tumor tissue images were taken at 40× magnification on processed slides (five regions per slide/liver) using a confocal microscope (Leica TCS SP5 II). ImageJ software (NIH, Bethesda, MD, USA) was used to count the positive cells/total cells and expressed as a percentage of apoptotic cells. Data were analyzed using GraphPad Prism 9.0.1 (licensed to the university). 

### 4.7. Tumor Burden in Mouse Liver

Images were taken at low magnification from H&E-stained liver slides from each liver and each animal, where matching and patching of all pictures were performed (Wax-it Histology Services Inc., Vancouver, BC, Canada) to obtain the whole liver section picture for pixel assessment. Pixel intensities were assigned to a specific channel for tumor vs. non-tumor area identification using the ImageJ software (NIH, Bethesda, MD, USA). Pixel color attributes on the intensity area were transferred into an Excel sheet for analysis using GraphPad prism 9.0.1. 

### 4.8. SCID Mouse Xenograft Model

Weekly mouse weight and in vivo imaging of tumor luminescence (IVIS Lumina XRMS Series 111 In vivo Imaging System, Waltham, MA, USA) were performed. At the time of animal surgery, the liver weight and maximum tumor size in millimeters (electronic digital caliper, TekDeals, San Marcos, CA, USA) were measured. Tumor slides per animal/group were assessed for different endpoints.

### 4.9. Confocal Microscopy Assessment on Immuno-Stained Cells/Liver Tissue

For in-vitro studies, immunocytochemistry of human HCC cell lines was performed as previously described [17]. Briefly, cells treated for four hours and untreated cells were plated on glass coverslips and allowed to reach 70% confluence. For α1-Na/K-ATPase and pSrc staining, cells were fixed (by adding ice-cold methanol for 10 min), permeabilized (0.05% Triton X-100), and then incubated with a monoclonal (α1-NKA, Millipore 05-369) or polyclonal antibody (to Tyr419 of Src kinase, Invitrogen 44-660G) overnight. The next day, the slides were incubated with secondary antibodies (Alexa Fluor 488 or 549 conjugate) and mounted on slides with Vectashield mounting media containing DAPI (Vector Laboratories, Inc.,Burlingame, CA, USA, H-1800). For Survivin and SMAC staining, cells were permeabilized and fixed using 4% Paraformaldehyde/0.05% Triton X-100, stained with polyclonal Survivin/SMAC antibodies (ab469/ab8115, Abcam, Cambridge, MA, USA), and then processed as described above. Images were taken at 63× magnification using a confocal microscope (Leica TCS SP5 II). The details of the antibodies used and their dilutions are listed in Appendix A.

For FoxO3 staining, the cells were fixed using 4% paraformaldehyde, permeabilized using 0.1% PBS-Tween 20 or 0.05% Triton X-100, and incubated with rabbit polyclonal anti-FOXO3A (Phospho S253, ab47285, Abcam, Cambridge, MA, USA) or FoxO3a monoclonal antibody (75D8, 2497S Cell Signaling Technologies, Danvers, MA, USA) overnight. The next day, the slides were incubated with secondary antibodies (Alexa Fluor 488 Conjugate) and mounted on slides with Vectashield mounting media containing DAPI (Vector Laboratories, Inc., H-1800). Images were taken at 63× magnification using a confocal microscope (Leica TCS SP5 II).

Immunohistochemical staining of human and mouse tissues was performed on formalin-fixed and paraffin-embedded liver and tumor tissue sections after deparaffinization and rehydration using xylene and graded ethanol exposure. In brief, liver and tumor tissue sections were subjected to antigen retrieval with 0.01 M citrate buffer (pH 6.0) and permeabilized with 0.1% Triton-X100 in PBS. Endogenous peroxidase activity and non-specific binding were blocked with H_2_O_2_ and protein blocks, respectively (kit-ab236469; Abcam, Cambridge, MA, USA). Sections were incubated overnight at 4 °C with primary antibodies against survivin/SMAC (Appendix A). After overnight incubation, sections were washed, incubated at RT in the HRP-conjugate (ab236469, Abcam, Cambridge, MA, USA), and visualized with DAB (3, 3’-diaminobenzidine tetrahydrochloride, ab236469, Abcam, Cambridge, MA, USA). The sections were then washed and counterstained with hematoxylin (H-3404, Vector Laboratories, Inc., Burlingame, CA, USA) and dehydrated to be cover slipped using VectaMount Permanent Mounting Medium (H-5000, Vector Laboratories, Inc., Burlingame, CA, USA). Images were taken at 40× magnification.

### 4.10. Quantitation of Protein Expression

Survivin and SMAC expression in the cytoplasm were quantitated using a described semi-quantitative technique that takes into consideration the intensity of positive staining and percentage of positive cells [72,73]. The color intensity of survivin and SMAC immunostaining was scored as follows: cell-free coloring (no staining), 0; light yellow (weak staining), 1; buffer (moderate staining), 2; brown (strong staining), 3. The percentage of positive cells was rated as follows: 0 < 5%; 1 = 5–25%; 2 = 25–50%; 3 = 50–75% and 4 > 75%. Scores for the percentage of positive cells and immunostaining intensities were multiplied to obtain an immunoreactive score (IRS) [72,73]. FoxO3 expression in cells was quantitated by measuring the integrated density of FoxO3 expression per unit area in the nucleus versus the cytoplasm using ImageJ software (NIH, Bethesda, MD, USA). The data were analyzed using GraphPad Prism 9.0.1.

### 4.11. Western Blotting

After the indicated treatment, cell lysates from ice-cold PBS-washed cells in RIPA buffer (pH = 7.4) were cleared by centrifugation (14,000 rpm for 15 min/4 °C), and supernatants were separated by SDS-PAGE and transferred to Protran nitrocellulose membranes (Thermo Fisher Scientific, Waltham, MA, USA). Blocked membranes were incubated with the same primary antibodies that were used for immunostaining each of the measured proteins and probed with the corresponding HRP-conjugated secondary antibodies ( Appendix A) for protein signals to be detected using the Pierce ECL kit (Thermo Fisher Scientific, Waltham, MA, USA). The blots were then quantitated using ImageJ software (NIH, Bethesda, MD) by measuring the integrated density of each of the blots after background subtraction and normalizing it against the housekeeping or total protein. Cell cytoplasmic and nuclear fractionation was performed using the NE-PER Nuclear and Cytoplasmic Extraction Reagents Kit (78335, Thermo-Fisher Scientific, Waltham, MA, USA) and Nuclear Extraction Kit (ab113474, Abcam, Cambridge, MA, USA), respectively. The success of nuclear and cytoplasmic extraction was confirmed via western blotting with the nuclear-specific protein LAMIN and cytoplasmic protein β-CATENIN, respectively. We used α-tubulin and β-actin as loading controls, which were ubiquitously and constitutively expressed in the two human HCC cell lines and were not affected by our treatment. The details of the antibodies used and their dilutions are listed inAppendix A.

### 4.12. Statistical Analysis

Results are presented as box-and-whisker plots. Data are presented as the median (central line), first and third quartiles (bottom and top of boxes, respectively), and whiskers (extreme values) from independent biological experiments. Differences between groups were determined by analysis of variance (ANOVA), followed by Tukey’s post hoc test or t-test using GraphPad Prism version 9.0.1 (GraphPad, San Diego, CA, USA). Statistical significance was set at *p* < 0.05. Statistical tests, sample size, and *p*-values are provided in the figure legends.

## Figures and Tables

**Figure 1 ijms-23-07359-f001:**
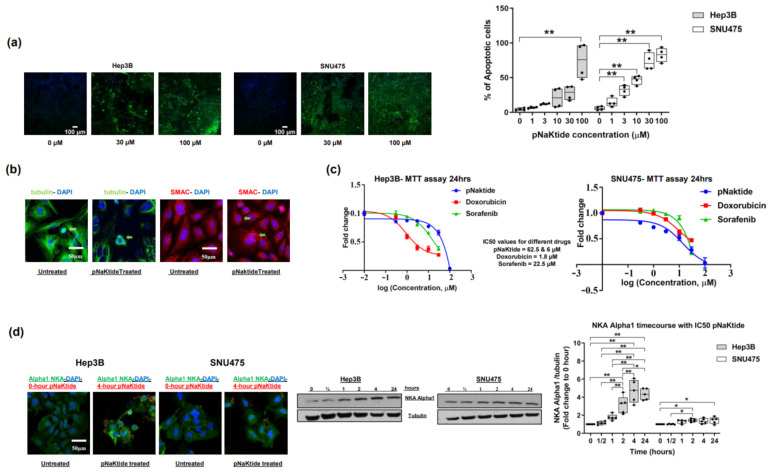
Effect of α1-Na/K-ATPase signalosome normalization on apoptosis and α1-Na/KATPase expression in Hep 3B and SNU475 cell lines. (**a**) Apoptotic activity was significantly increased in pNaKtide-treated human HCC cell lines in a dose-dependent manner (positive cells were assayed by TUNEL, and shown as box-whisker plots, *n* = 4. ** *p* < 0.01, by ANOVA and Tukey’s Post hoc test). (**b**) Mitotic catastrophe in SNU475 cell lines following α1-NKA signalosome normalization. Notice the presence of multiple micronuclei and aneuploidy in the treated cells (arrows) compared with the untreated group. (**c**) Cell toxicity of IC50-pNaKtide in Hep3B and SNU475 human HCC cell lines (62.5 µM and 6 µM, respectively) compared with sorafenib and doxorubicin (*n* = 5). (**d**) Confocal microscopy images of the effect of pNaKtide on α1-subunit expression and a 24 h time course of α1-subunit expression by western blotting in two human HCC cell lines exposed to IC50-pNaKtide (*n* = 5, * *p* < 0.05, ** *p* < 0.01).

**Figure 2 ijms-23-07359-f002:**
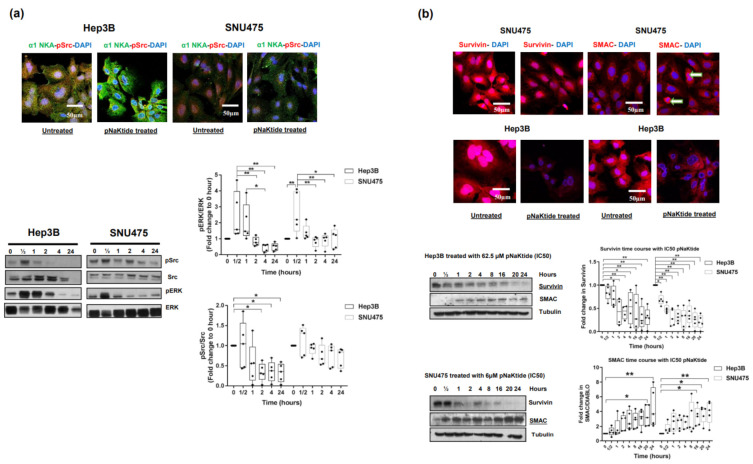
Effect of α1-Na/K-ATPase signalosome normalization on pSrc, Survivin and SMAC protein expressions in Hep 3B and SNU475 cell lines. (**a**) Confocal microscopy images of the effect of pNaKtide on Src expression and a 24 h time course by western blotting in two human HCC cell lines. Box-whisker plots are showing a fold change in each protein relative to baseline at 0 h (*n* = 5, * *p* < 0.05, ** *p* < 0.01). (**b**) Confocal microscopy images of survivin and SMAC protein expressions in two human HCC cell lines IC50-pNaKtide treated vs. non-treated cells and a 24 h time course analysis by western blotting. There was a significant progressive decrease in survivin expression with concomitant increase in SMAC levels over time (*n* = 5, * *p* < 0.05, ** *p* < 0.01). An earlier effect was noted for survivin (significantly decreased for Hep3B at 2 h and for SNU475 cells at 30 min) when compared with SMAC expression (significantly increased for Hep3B at 20 h and for SNU475 cells at 16 h).

**Figure 3 ijms-23-07359-f003:**
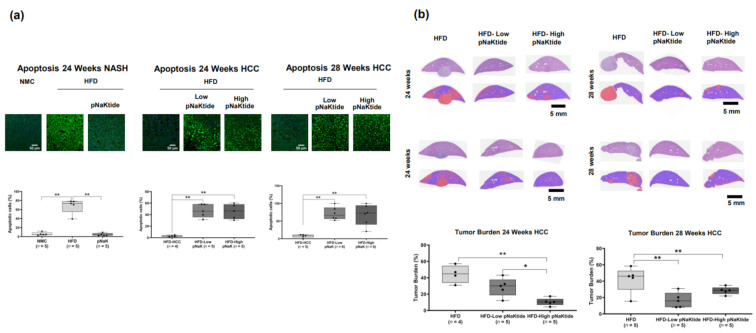
Effect of α1-Na/K-ATPase signalosome normalization in-vivo on apoptosis and tumor burden. (**a**) The apoptotic activity of livers from animals with NASH was increased significantly when compared with controls (normal mouse chow group) and pNaKtide abrogate such activity on livers from treated animals. In contrast, the apoptotic activity was significantly increased in the NASH-HCC mouse model by an apoptotic ‘switch’ promoted by pNaKtide (*n* = 5–6 mice for each group, ** *p* < 0.01 by ANOVA and Tukey’s Post hoc test, on TUNEL stained liver slides). (**b**) There was a significant decrease in the tumor burden of livers treated with pNaKtide from NASH related HCC at 24 and 28 weeks when compared with livers from untreated animals (*n* = 5–6 mice for each group, * *p* < 0.05, ** *p* < 0.01). The effect of α1-NKA signalosome normalization was dose dependent at the 24-week arm of the experiment.

**Figure 4 ijms-23-07359-f004:**
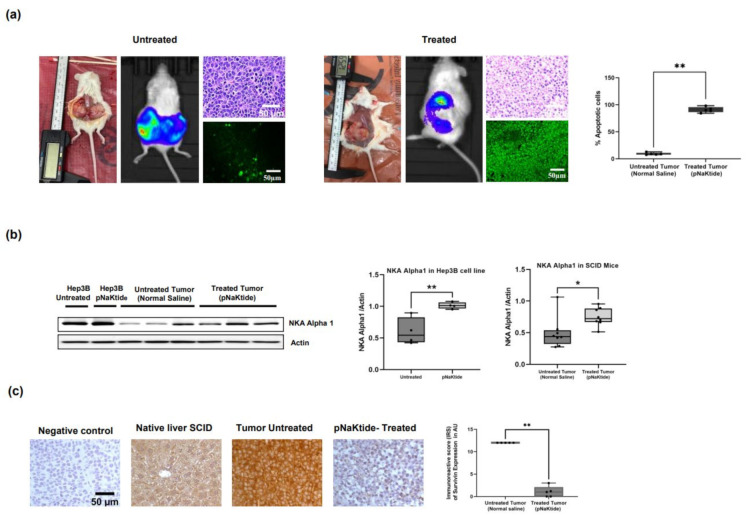
Effect of α1-Na/K-ATPase signalosome normalization on SCID mice xenograft tumor cells. (**a**) Representative images of tumors from untreated vs. pNaKtide treated SCID mice xenograft tumors. Tumor sizes were assessed via the use of video-IVS method and electronic calipers measurement. Representative images of H&E staining of tumor sections from untreated and treated mice are also shown (**top**). Additionally, representative confocal images of apoptosis in untreated vs. pNaKtide treated tumor are shown (**bottom**). Notice, the difference in tumor cells histology and heightened apoptosis in the treated tumor vs. untreated tumor (*n* = 5. ** *p* < 0.01, *t*-test). (**b**) Increase α1-Na/KATPase expression in the xenograft tumors of the treated mice compared with the untreated mice (*n* = 8, * *p* < 0.05, ** *p* < 0.01, *t*-test). (**c**) Survivin expression in pNaKtide-treated tumors vs. untreated tumors. Survivin expression was significantly down regulated in the treated mice tumors in comparison with the untreated mice (*n* = 5. ** *p* < 0.01, *t*-test).

**Figure 5 ijms-23-07359-f005:**
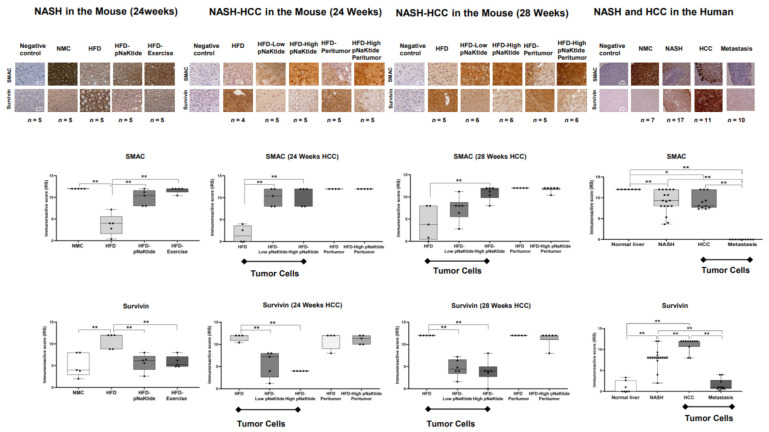
Effect of α1-Na/K-ATPase signalosome normalization on survivin and SMAC proteins expression in NASH and NASH-HCC mouse models and human liver tissues. There was a significant down-regulation of survivin and upregulation of SMAC expressions in tumor cells from treated vs. non-treated animals at 24 and 28 weeks (** *p* < 0.01, by ANOVA and Tukey’s post hoc test). There were no significant changes in protein expressions in the peritumor cells of mice in the HFD groups vs. high pNaKtide-treated groups in the NASH-HCC mice model (24 and 28 weeks). In human tissues, the findings were consistent with the in-vitro and in-vivo findings for both survivin and SMAC proteins. There was a significant upregulation of survivin expression in livers from patients with NASH and HCC when compared with livers from normal subjects and patients with liver metastases (** *p* < 0.01). On the contrary, SMAC expression significantly decreased in livers from patients with NASH, HCC, and liver metastases when compared with livers from the normal subjects (* *p* < 0.05; ** *p* < 0.01).

**Figure 6 ijms-23-07359-f006:**
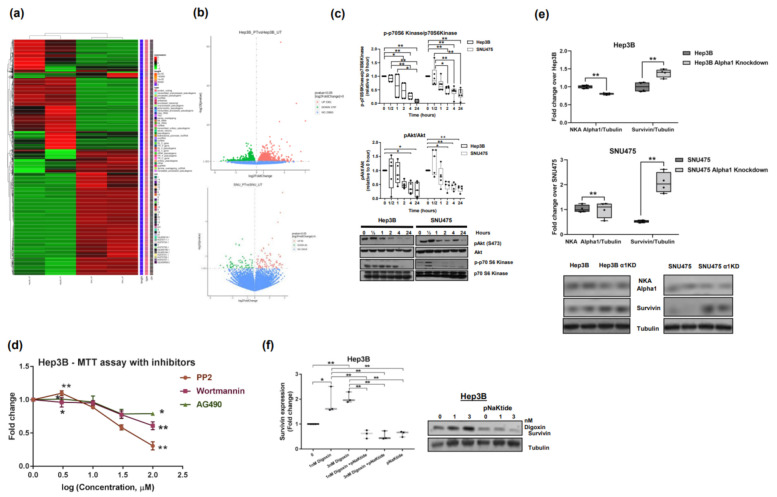
α1-NKA Signalosome. (**a**,**b**) Heat map and volcano plots showing differential gene expressions in pNaKtide-treated vs. -untreated human HCC cell lines (*n* = 5). Significant changes were observed in genes that play key roles in PI3K → Akt pathway. Red color indicates genes with high expression levels, green color indicates genes with low expression levels and blue indicates unchanged genes (Novogene, Sacramento, CA, USA). (**c**) Data showing significant increase in the activation of phospho-Akt and phospho-S6k1 in untreated human HCC cell lines when compared with pNaKtide-treated cells (* *p* < 0.05, ** *p* < 0.01, by ANOVA and Tukey’s post hoc test, *n* = 4–6). (**d**) MTT assay of the PI3K/Akt pathway inhibitors, wortmannin, PP2 and AG490, in the Hep3B cell line. Observe the significant arrest of cell proliferation following drug administration. (**e**) Survivin over-expression in α1-NKA Knockdown (siRNA) cells from two human HCC cell lines (**p* < 0.05, ***p* < 0.01). Representative Western blots showing a1 Na/KATPase, surviving, and tubulin expression in control siRNA and targeted iRNA transfected cells are shown. (**f**) Digoxin induced upregulation of survivin expression in the Hep3B cell line. Note an abrogation of digoxin effect on cells exposed to pNaKtide (* *p* < 0.05, ** *p* < 0.01).

**Figure 7 ijms-23-07359-f007:**
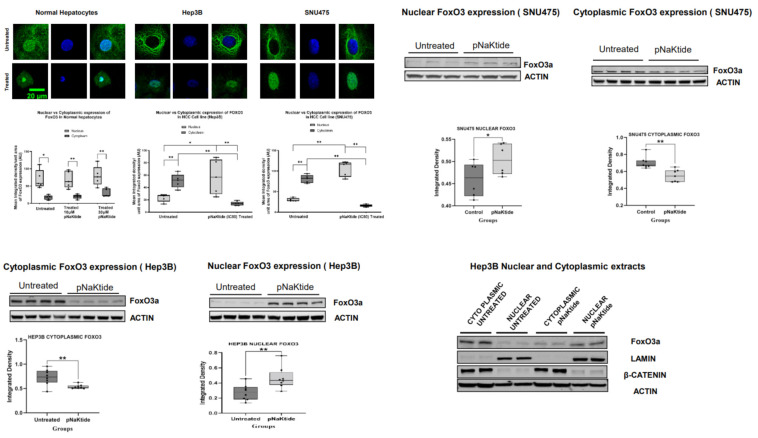
Effect of α1-Na/K-ATPase signalosome normalization on nuclear/cytoplasmic expression of FoxO3. Nuclear/cytoplasmic expression of FoxO3 in pNaKtide-treated vs. -untreated human normal hepatocytes and HCC cell lines cells (* *p* < 0.05, ** *p* < 0.01, paired *t*-test, *n* = 5). Notice the significant nuclear localization and low cytoplasmic expression of FoxO3 in normal hepatocytes and in pNaKtide-treated HCC cells compared with the untreated cells. Additionally, note the expression of nuclear LAMIN and cytoplasmic β-CATENIN in western blots, confirming success of nuclear/cytoplasmic extraction.

**Figure 8 ijms-23-07359-f008:**
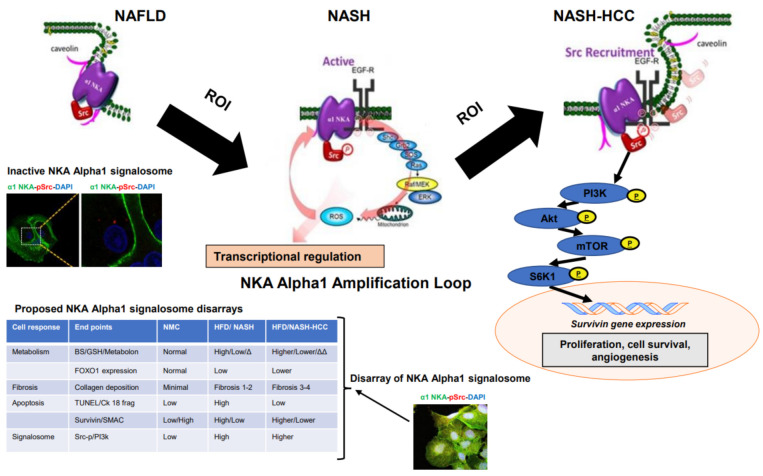
Proposed schematic diagram of activated α1-NKA Signalosome mediated NASH-related HCC malignancy. At the onset of NAFLD, the signalosome is inactive but as NAFLD progresses there is generation of ROI, resulting in the activation of the signalosome and induction of NASH. Continuous activation of the signalosome by ROI, coupled with the underlying NASH microenvironment results in the amplification of ROI production via the EGFR-Ras/Raf/MEK/ERK cascade, turning into a vicious cycle that intensifies the activation of the α1-NKA/Src signalosome which concludes in genesis of liver malignancy by survivin overexpression. Survivin induces carcinogenesis by inhibiting apoptosis, and heightening cell proliferation and tumor angiogenesis. The chart also displays certain proteins and metabolic and morphological disarray that drive activated α1-NKA signalosome-mediated hepatocarcinogenesis. (Note: EGFR, Ras/Raf/MEK/ERK pathway symbols adapted from [9] Cui and Xie, 2017; *Molecules).*

## Data Availability

Not applicable.

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
