# Peer review of "Tumor-Suppressor Role of the α1-Na/K-ATPase Signalosome in NASH Related Hepatocellular Carcinoma"

_ijms, 2022, doi:10.3390/ijms23137359_

Round 1

Reviewer 1 Report

Udoh UA et al. presented the tumor-suppressor role of the alpha1-NK/K-ATPase signalosome in NASH related HCC. They also showed that dysregulation of this molecule in HCC is associated with concomitant downregulation of the alpha1-NKA subunit, upregulation of the anti-apoptotic protein survivin, and the downregulation of the pro-apoptotic SMAC protein expression, promoting a cell fate switch from apoptosis to mitosis. These findings were carefully proved by many experiments and are hopeful and interesting, but still have some concerns below.

Major

1.    In NASH-HCC, pNaKtide treatment showed a significantly lower tumor load (Figure 3b). This was seen in a dose-dependent manner at 24 weeks, but not at 28 weeks. High-dose of pNaKtide oppositely increased tumor burden of livers. There were no significant differences but I have some concerns about treatment of high-dose pNaKtide. Could you please explain and speculate some mechanisms.

2.    The authors explain the schematic diagram of activated alpha1-NKA signalosome NASH-related HCC malignancy in Figure 8. Not only PI3k pathway but STAT pathway is related to survivin expression. The authors have never examined the activation of STAT3 in alpha1-NKA signaling. They have to exclude the activation of STAT under treatment with pNaKtide. 

3.    The authors explained that pNaKtide attenuated the nuclear-to-cytoplasmic export of FoxO3 protein and favored its nuclear localization. They should show the way how to export FoxO3, and keep it in nucleus.

4.    The authors concluded survivin is a key protein that drives the progression of NASH to HCC. Digoxin is a specific ligand of NKA and it increased the expression of survivin. So we can easily speculate NASH patients should not be treated with digoxin. Is it right? Are there some papers to support this speculation?

Author Response

 Responses in the attachment.

Reviewer 2 Report

In this manuscript by Udoh et al, the authors highlight the function of α1-Na/K-ATPase signalosome in NASH related heptocellular carcinoma. The manuscript is well written and data clearly presented. However, prior to publication, can the authors please address the following questions?

1. Why is Supplementary Figure S2c introduced before Figure S1 and S2a?

2. Please provide the tubulin loading for the western blots in Figure 6e.

3. In Figure 6e, was a control siRNA used to compare the NKA alpha 1 siRNA knockdown? A control siRNA must be used to compare the siRNA knockdown and can this be clearly explained in the Figure and legend?

4. In Figure 6f, how was the survivin expression calculated? Please provide the loading control for this western blot.

5. Can the western blots in Figure 7 be cleared labelled with the proteins that were probed for?

Author Response

Responses in the attachment. please see attachment

Round 2

Reviewer 2 Report

The rebuttal letter has sufficiently addressed my concerns. I recommend acceptance of this manuscript.